# Dietary Profile and Nutritional Status of the Roma Population Living in Segregated Colonies in Northeast Hungary

**DOI:** 10.3390/nu12092836

**Published:** 2020-09-16

**Authors:** Erand Llanaj, Ferenc Vincze, Zsigmond Kósa, János Sándor, Judit Diószegi, Róza Ádány

**Affiliations:** 1Department of Public Health and Epidemiology, Faculty of Medicine, University of Debrecen, Kassai street 26/B, H-4028 Debrecen, Hungary; erand.llanaj@med.unideb.hu (E.L.); vincze.ferenc@med.unideb.hu (F.V.); sandor.janos@med.unideb.hu (J.S.); 2Doctoral School of Health Sciences, University of Debrecen, Kassai street 26/B, H-4028 Debrecen, Hungary; 3Department of Methodology for Health Visitors and Public Health, Faculty of Health, University of Debrecen, Sóstói street 2–4, H-4400 Nyíregyháza, Hungary; kosa.zsigmond@foh.unideb.hu; 4MTA-DE-Public Health Research Group, University of Debrecen, Kassai street 26/B, H-4028 Debrecen, Hungary; dioszegi.judit@med.unideb.hu

**Keywords:** nutrition, Roma, Hungary, diet, nutritional status, dietary intake, 24-h recall

## Abstract

Nutrition studies among Roma are scarce and to date no quantified dietary data are available. This report provides, for the first time, quantified dietary data and comprehensive anthropometric information for the Hungarian Roma (HR) population, with Hungarian general (HG) adults as reference. Data were obtained from a complex comparative health survey, involving 387 and 410 subjects of HR and HG populations, respectively. Using corporal measurements, body composition indicators were constructed, while daily nutrient intakes were evaluated in comparison with internationally accepted guidelines on nutrient requirements and recommended intakes. Associations between Roma ethnicity and nutrient intakes, as well as odds of achieving dietary recommendations were explored using regression models, adjusted for relevant covariates (i.e., age, gender, education, marital status and perceived financial status). Results showed occasional differences for selected nutrient intakes between the groups, with HR’s intake being less favorable. Total fat intake, predominantly animal-sourced, exceeded recommendations among HR (36.1 g, 95% confidence interval (CI): 35.2–37.0) and was not dissimilar to HG group (37.1 g, 95% CI: 36.3–38.0). Sodium intake among HR was significantly lower (5094.4 mg, 95% CI: 4866.0–5322.8) compared to HG (5644.0 mg, 95% CI: 5351.9–5936.0), but significantly greater than recommended intake in both groups. HR had greater estimated body fatness (25.6–35.1%) and higher average body mass index (BMI, 27.7 kg/m^2^, 95% CI: 26.9–28.4), compared to HG. In addition, HR had lower odds of achieving dietary recommendations (odds ratio (OR) = 0.81, 95% CI: 0.67–0.97, *p* < 0.05). Findings warrant further research, while highlighting the importance of establishing and integrating Roma nutrition into national surveillance and monitoring systems for key dietary risk factors.

## 1. Introduction

The Roma population constitutes the largest ethnic minority in Europe (estimated to be between 10–12 million) [1] and has been a major focus of ethnicity-based studies in past decades [2,3]. Historical migration has led to their dispersion over many European countries [4,5,6] and they have been subjected to disadvantaged living conditions, food insecurity, high-risk health-related behaviors, as well as discrimination, stigmatization and barriers to accessing health services [7,8,9,10,11]. These exposures have resulted in a particularly poor health status indicated by the high prevalence of chronic morbidity, particularly diet-related non-communicable diseases (NCDs) [12,13,14,15,16,17,18], as well as an estimated life expectancy of almost a decade less, compared to the general population in Europe [16,19,20].

In addition, survey results from different European countries, attempting to elucidate dietary aspects of Roma, have indicated detrimental dietary patterns, linked to the common hallmarks of diet-related NCDs, namely inadequate and infrequent consumption of fruits, vegetables [21,22,23,24,25,26,27,28,29,30,31,32,33] and dairy products [25,27,28,29,34], unfavorable consumption of fast foods [35,36,37,38], animal fats [30,33,38,39], sugar-sweetened beverages [24,25,30,37], as well as confectionery [30]. Although these studies have described food choices and eating habits, several questions on dietary intake profile still remain unanswered. Apart from a recent attempt to qualify dietary intake of Roma [31], nutrient intake patterns and data among Roma are not available yet in Hungary or elsewhere. Such information would be very useful, not only in recognizing how diet may be related to health, but also in determining which specific aspect(s) of diet or nutrient(s) should be targeted, when designing and/or implementing preventive interventions among Roma, i.e., identifying effective ways of intervening to reduce health inequalities.

The current representation of Roma in Hungary is over 8% of the total population and it is expected to further increase, due to the significantly higher fertility rate among Roma compared to that of the majority population [40,41]. The spatial patterns of Roma show their intense segregation and peripheralization mainly in Northeast Hungary and South Transdanubia [41]. Within less than a decade, there has been a robust increase in the prevalence of obesity, in both sexes, among younger Hungarian Roma (HR) and the same trend holds true for middle-aged women [3]. Since these new nutrition-related challenges are arising among HR, it is necessary to have a clear understanding of the dietary profile and nutritional status of this underserved ethnic group, to better address malnutrition challenges, in all their forms, as well as commit to the Rome Declaration on Nutrition’s common vision to eradicate all forms of malnutrition, in the framework of the UN Decade of Action on Nutrition (2016–2025).

The dietary profile of the Hungarian adult population has been extensively investigated and is largely characterized by the intake of immoderate amount of fats, meats (mainly processed), salt and sugar [42,43]. However, dietary intake and nutrient patterns of HR adults (nor Roma adults in other countries) have never been quantified. We hypothesized that the more unfavorable health status of Roma in comparison with the general population is partly related to their unhealthier dietary profile and nutritional status. Hence, examining the dietary profile and nutritional characteristics of Roma, particularly those living in highly deprived segregated settlements, can provide an enhanced understanding on their nutrition and its potential interrelationship (s) with their health, which in turn can help to inform and guide policies and decision-making

To test our hypothesis, we used data obtained from our Complex Health (Interview and Examination) Survey [41], with the intention of investigating the nutritional characteristics and dietary behavior of this ethnic group living in segregated colonies in Northeast Hungary where they are primarily concentrated. To the best of our knowledge, this is the first study aiming at providing quantified, extensive dietary intake data and comprehensive nutritional estimates for Roma adults, in comparison with not only general population’s data as reference, but with data specified in established guidelines/recommendations for healthy nutrition as well.

## 2. Materials and Methods

### 2.1. Study Design and Sampling

All data used in this analysis were obtained in a cross-sectional survey carried out between May and August 2018, as a three-pillar (i.e., questionnaire-based, physical examination and laboratory examination) complex (i.e., health behavior and examination) survey. Details of sampling and data collection and management are thoroughly described elsewhere [44]. In brief, individuals aged 20 to 64 years, were selected randomly, to be representative of the adult HR population living in segregated colonies of North-East Hungary (Hajdú-Bihar and Szabolcs-Szatmár-Bereg counties), where a great proportion of the HR population resides [41], as well as that of the HG population living in the same counties. In addition to the demographic, anthropometric, health behavior, physical and laboratory data collection in the three pillars of the survey two 24-h recalls were also obtained to quantify dietary intake. The intended sample size was 500 participants for both study groups, but the final study sample, with full recall data, included 797 participants, of whom 410 subjects of the HG and 387 individuals of the HR population.

#### 2.1.1. Hungarian Roma Sample

Segregated colonies exceeding 100 inhabitants were identified previously by Roma field workers and ethnicity of the colony population was assessed by self-declaration, as described elsewhere [45]. Roma participants were enrolled based on a stratified multistep sampling method in two counties of Northeast Hungary (i.e., Hajdú-Bihar and Szabolcs-Szatmár-Bereg), where the majority of Roma colonies reside. After verification of a previously created database, twenty colonies were randomly chosen, and twenty-five households were randomly drawn from each colony. Individuals aged 20 to 64 years were identified in each household and eventually one person was selected by random table. These persons were interviewed face-to-face at the respondent’s household by Roma university students with the support of the local Roma self-government under the supervision of public health coordinators.

#### 2.1.2. Hungarian General Reference Sample

The Hungarian reference sample was derived from the General Practitioners’ Morbidity Sentinel Stations Program (GPMSSP), which is a population-based registry designed to monitor NCDs of high public health importance, operating since 1998 in Hungary. Detailed description of GPMSSP is presented elsewhere [46]. Individuals, aged 20 to 64 years, living in private households and registered by GPs in the same aforementioned counties, were randomly chosen from the GPMSSP registry. The original plan of the study was to involve 25 randomly selected individuals from 20 GP practices in these counties, but because of two GPs refusing to participate, the final sample was reduced to 450 participants from the practices of 18 GPs. In the present study individuals with full recall data were included from both groups, but subjects with implausibly low or high intake data were excluded.

#### 2.1.3. Socio-Demographic Data

Socio-demographic characteristics of participants were defined by using data obtained in the complex health survey, including age, sex, educational level (‘elementary’, ‘secondary’, ‘vocational training’ and ‘university degree’), self-reported perceived financial status (on a Likert scale: ‘very good’, ‘good’, ‘fair’, ‘challenging’ and ‘very challenging’) and economic activity (‘full-time employment’, ‘part-time employment’, ‘student’, ‘unemployed’, ‘retired’ and ‘ill-health retirement’-the latter including subjects that were unable to work any longer due to illness or disability).

#### 2.1.4. Dietary and Anthropometric Data Collection and Quality Appraisal

Dietary intake data were obtained in the case of each participant through a double (i.e., one non-consecutive weekday [excluding Monday] and one weekend day) interviewer-assisted multiple-pass 24-h dietary recall protocol developed and validated in our previous study [47]. Instructions on recording food intake were administered on a step-by-step basis, elaborating all aspects of qualification and quantification, usage of illustrated accompanying booklet and coding procedure. For more details see section ‘Dietary data’ in the Appendix A. Briefly, the consecutive steps used to determine food intake were the following: (i) listing of foods consumed by the respondent during a 24-h period, including one weekday and one weekend day, before the interview, (ii) additional recall of nine categories of foods that are often forgotten, such as non-alcoholic and alcoholic beverages, sweets, savory snacks, fruits, vegetables, cheeses, bread and rolls, etc., (iii) assessment of meal time and occasion, (iv) detailed description of each item reported, and (v) a final opportunity to amend or recall any other unreported item.

All subjects were eligible to report their intake if the weekday and the weekend day were considered ‘typical’, which meant that: (i) intake represented what they usually consume, (ii) there was no special event (birthday, party, etc.) on the day assessed, (iii) no disease was diagnosed prior or during the days recorded, (iv) nutritional supplements of any kind were taken and (v) a diet regime of any kind was applied. If one of these conditions was not satisfied, subjects were not eligible and hence the recall was not recorded. The two dietary recalls were administered to all survey participants.

Generalized obesity and obesity classes were estimated based on body mass index (BMI) according to World Health Organization (WHO) criteria (i.e., obesity class I (30.0–34.9 kg/m^2^), class II (35.0–39.9 kg/m^2^) and class III (above 40 kg/m^2^) [48]. Abdominal obesity was determined using waist circumference (WC) standards defined for females and males by the International Diabetes Federation (IDF) for European (IDF_EURO_) population [49]. The waist-to-height ratio (WHtR) index [50] was also calculated, as it has received attention in the scientific literature for being strongly associated with metabolic syndrome and several related NCDs [51,52,53] regardless of sex, age, or ethnic group [54]. In addition, anthropometric indices that estimate percentage of body fat (PBF) were defined, using four different anthropometric equations for estimating PBF [55,56,57,58]. These equations were considered suitable and potential alternatives in estimating whole-body fat percentage in subjects 20 years of age and older, independent of ethnic background and sex [59].

Eventually, the proportion of those with metabolically healthy obesity (MHO) was determined among subjects suffering from obesity (i.e., BMI ≥30 kg/m^2^). Since, there is no universally accepted standard for defining MHO, and there are more than 30 different definitions in the literature [60], MHO was defined in our study if participants fulfilled the following criteria: (1) no diagnosed pre-existing cardiometabolic diseases, (2) a healthy cardiometabolic blood profile (i.e., fasting triglycerides (TG) <95 mg/dL, High-Density Lipoproteins—Cholesterol (HDL-C) ≥40 mg/dL (men) and ≥50 mg/dL (women) and fasting glucose <100 mg/dL,) and (3) normal BP (i.e., blood pressure <130/85 mmHg). After applying these criteria, we used four different approaches to further determine proportion of MHO [61,62,63,64].

### 2.2. Data Analysis

Dietary data were processed with NutriComp Étrend ver. 3.03 (https://www.nutricomp.hu/) software which has been used previously in Hungarian Diet and Nutritional Status surveys [42,43], as it contains detailed food composition information on 1328 food items and 1823 recipes. Additional recipes were created, or existing ones were modified in the software’s database, if a food item was missing or had additional/fewer ingredients accordingly. The software converts inputs of food and drinks intakes, into quantified macro-and micro-nutrient intakes and creates a mean of the two dietary recalls.

Comparisons of characteristics between the study groups were made using chi-square (χ^2^) test for categorical variables and *t*-test or analysis of variance (ANOVA) for continuous variables, as appropriate. Data were presented as means accompanied by 95% confidence intervals (95% CI), where possible, or as percentages. Additional statistical tests were used (trend analysis) and all assumptions for normality were checked. Multiple linear regression analyses were performed for exploring the associations between Roma ethnicity and nutrient intakes.

Regression coefficients were adjusted for the respondent’s age, gender, and education, marital and perceived financial status. Beta coefficients (*β*) and their corresponding 95% CI were calculated for explanatory parameters. In linear regression analyses, outcome variables with a non-normal distribution were normalized by Box–Cox transformation [65]. Furthermore, binary outcomes were created based on international nutrient intake recommendations and a multiple logistic regression was used to compute the odds of achieving nutrient intake recommendations for the HR sample, with HG group as a reference, while adjusting for all relevant covariates (i.e., age, gender, and education, marital and perceived financial status). Results were presented as forest plots and an estimated overall quantitative value for the combined odds was shown for both fixed and random effect models [66]. Recommended intake values were obtained from WHO’s official guidelines on diet, nutrition and the prevention of chronic diseases [67], as well as joint WHO/Food and Agriculture Organization (FAO) guidelines on fats and fatty acids [68], vitamins and minerals [69], sodium [70] and potassium [71]. In addition, recommended intake targets derived from the Dietary Approaches to Stop Hypertension (DASH) [72], the EAT-Lancet reference diet [73] and European Food Safety Authority (EFSA) dietary reference values [74] were considered for benchmarking. Comparisons of intakes with the recommended/reference intakes have been made.

Statistical analyses were performed with IBM SPSS Statistics for Windows, Version 21.0 (IBM Corp., Armonk, NY, USA) and R Statistics (RStudio, Boston, MA, USA). Results were reported in accordance with STROBE (Strengthening the Reporting of Observational studies in Epidemiology) extension for nutrition and dietary assessment [75].

### 2.3. Research Ethics

Approval for the research protocols and methodology was provided by the Ethical Committee of the Hungarian Scientific Council on Health (61327–2017/EKU). Participants gave their written informed consent in each study population in accordance with the Declaration of Helsinki and the Science Ethics Code of The Hungarian Academy of Sciences.

## 3. Results

The final study sample comprised 797 participants, 410 in the HG and 387 in the HR population, with response rates of 91.0% and 77.4%, respectively. Those who had implausibly low or high estimated caloric intake (<800 or >4500 kcal/day for males, <700 or >3500 kcal/day for females [47,76,77]), were excluded, i.e., 51 HG subjects (4 and 47 for implausibly low intake and high intake, respectively) and 43 HR subjects (4 and 39 for implausibly low intake and high intake, respectively), thus 703 participants aged 20 to 64 years (i.e., 359 HG and 344 HR) were included in the final analysis. For more details on the characteristics of excluded subjects see Appendix A.

### 3.1. Sociodemographic and Anthropometric Characteristics of Participants

Both populations had a higher representation of females and HR status was associated with lower education higher unemployment levels and more perceived financial difficulties (Table 1). All these associations were statistically significant.

Average WC and BMI values were not significantly different between the two groups (they fell within the range of overweight status in both populations), but the distribution of BMI categories showed significant difference. Abdominal obesity, based on IDF_EURO,_ was significantly associated with ethnicity only among men (Table 2). Both underweight and obesity were more prevalent among HR. WH*t*R was significantly higher among HR, while females, in both groups, had significantly higher PBF compared to males, regardless of ethnicity.

There were significant differences between the two groups on PBF, regardless of the method used to estimate it—with HR having consistently higher estimates of PBF than HG. The representation of MHO subjects was consistently, but not significantly lower in the HR group compared to HG, regardless of the classifying criteria applied.

### 3.2. Dietary Intake Patterns

Energy intake estimation was found to be not significantly different between HG and HR. Males had higher energy intake compared to females, regardless of ethnicity (Table 3), but this difference was found statistically not significant.

Total carbohydrate daily intake, as energy percentage, was significantly higher among HR, but still significantly lower than the recommended range in both groups (Table 4). Sugar intake did not differ significantly between HR and HG, but it was significantly higher than the recommended daily intake of 10% of total energy intake (10%E), and even higher than the 5%E intake recommended by WHO for additional health benefits. Total dietary fiber intake was much lower than the recommended daily amount for both groups.

Total dietary protein intake was significantly higher compared to recommended intake ranges among HG, but not among HR. Neither animal-based nor plant-based protein intake were significantly different between groups. Total and essential amino acid consumption was significantly lower in the HR sample compared to the HG group.

There were no significant differences by type and source of fat between the two groups, but there were significantly higher intakes (in both groups) compared to the established dietary recommendations. Polyunsaturated fatty acids’ (PUFAs) intake was significantly lower among HR, but still within the recommended range and both groups were characterized by an excessive intake of SFAs. Cholesterol intake was very high compared to the reference limit in both groups, both as absolute intake and as adjusted value (i.e., mg/1000 kcal); while intake of beneficial fatty acids, such omega-3 fatty acids and alpha-linolenic acid, were very low, particularly among HR. Omega-6 intake was higher than the upper value of the recommended range among HG and significantly lower, but at the upper value among HR.

In case of minerals and trace elements, sodium intake in both groups was exceedingly higher compared to established international dietary recommendations (Table 5), while potassium and magnesium intakes were below the recommended intake, independently of the criteria used. Intake values were not significantly different between the two groups. HR had consistently lower vitamin intake—particularly B vitamins—compared to recommendations. For more in-depth comparisons of nutrient intakes between males and females see Appendix A.

When examining the odds of the HR participants achieving the recommended daily nutrient intake ranges/values, compared to HG population as a reference, results showed that HR was less likely to achieve recommended intake targets, compared to HG (Figure 1).

The odds were significantly lower for HR in general, regardless of the model accounted for (fixed and random effects). Vitamin D had a wide 95% CI range, as there was a very limited number of participants who achieved recommended intake for this micronutrient.

## 4. Discussion

Recently race- and ethnicity-based health disparities have become a central focus of public health research, practice, and policy, as a growing body of evidence shows a strong association between racial/ethnic and socio-economic disparities with healthy dietary and nutrient patterns [79,80,81]. Diet-related disparities play an important role and exist in the form of “differences in dietary intake, dietary behaviors, and dietary patterns in different segments of the population, resulting in poorer dietary quality and inferior health outcomes for certain groups and an unequal burden in terms of disease incidence, morbidity, mortality, survival, and quality of life” [82]. Since diet and nutrition are closely related to a number of noncommunicable diseases, there is growing interest in characterizing the association between dietary and nutrient intake in specific disadvantaged minority population groups, such a Roma. In Europe, the Roma population constitutes the largest ethnic minority (estimated to be between 10–12 million) [1] and has been a major focus of ethnicity-based studies in past decades [2,3]. The poor living conditions in which some Roma people live, frequently on the outskirts of towns and villages and in substandard settlements, allow relatively straightforward identification of locations in which Roma people are concentrated [45]. This study has taken advantage of this opportunity, by sampling HR participants in Northeastern Hungary, where the Roma population is greatest and in identified settlements, in which the population was almost exclusively Roma.

To our knowledge, this is the first study designed to characterize and examine selected health-relevant nutrients estimates and anthropometric parameters, as well as the dietary intake patterns and profile of the Hungarian Roma, with reference to the Hungarian general population, based on data derived from our complex health survey [3]. Findings indicate overall poor dietary patterns for both groups, with inadequate dietary composition and anthropometric status estimates, not strongly different, but occasionally worse among HR. The dietary profile of HR participants could be characterized by lower odds for achieving established dietary recommendations, highlighting the need for additional public health initiatives to translate nutritional data into efficacious preventive interventions to lower risk of malnutrition in all its forms, as well as diet-related NCDs risk.

With regard to nutritional status, HR appear to be particularly affected by malnutrition in many forms, with less favorable estimates of body composition coupled with greater perceived financial challenges and higher unemployment rates—factors which may affect access to better nutrition and dietary quality. Although statistical differences could not be detected for some anthropometric indices, estimates of body fatness were significantly and consistently (criteria-wise) higher among HR, indicating less healthy body composition compared to HG. Although not significantly different from HG, consistently lower MHO was shown among HR according to different classification criteria. Such results need to be confirmed via direct body composition measurements, but currently these findings are in line with results from recent analysis of two paired health surveys, where the distribution of BMI was shown to have significantly worsened among younger HR (in both sexes) between 2004 and 2015, with obesity becoming significantly more frequent [3]. In addition, Roma had higher rates of underweight compared to HG.

With regards to nutrient patterns and intake, dietary fat composition among the study participants, was substandard considering the representation of beneficial fatty acids, such as PUFAs, omega-3 fatty acids and alpha-linolenic acid, particularly among HR. SFAs and cholesterol intake were excessively high in comparison with the recommended intake, with no significant differences between groups. These results are consistent with recent WHO estimates, that show the adult population in Hungary with an estimated 11.8% of their total calorie intake coming from SFAs [83]. It is reasonable to assume that such high SFAs’ intakes can be partially explained by the traditional consumption of meat and SFAs-rich products, such as lard, tallow, cold cuts and sausages among the Hungarian population [84]. The current nutritional discourse and best dietary guidelines put no longer an emphasis on the reduction in total fat intake, but rather call for optimization of fat types in the diet, and specifically reduced intake of SFAs and trans-fats [85]. Therefore, given our results and the current evidence, dietary guidance should focus on optimizing dietary fat sources.

Similarly, a recent survey in South Bohemia Region (*Czech Republic*) [31], attempting to ascertain differences in eating habits between Roma and the majority population, based on qualification of major nutrient intake estimates, showed that Roma exceeded reference values for energy intake (as calculated in relation to gender, age and physical activity) and the proportion of fats (i.e., 34.9%E) in their diet was higher than the nutritional recommendations.

Furthermore, high intake of animal-based proteins suggests a high meat and meat products consumption, supported by results of previous research on HG [86]. A higher proportion of plant-based proteins compared to animal-based proteins (regardless of type of meat: white or red meat) in diet has been shown to improve cardiometabolic profiles and reduce cardiovascular disease risk, on the basis of plasma lipid and lipoprotein effects [87]. The opposite, i.e., longitudinal higher intake of animal-based proteins, has been associated with increased risk of insulin resistance and prediabetes and type 2 diabetes (T2D) [88]. Both groups had a greater than one animal to plant protein ratio and HG had significantly higher compared to HR. This result cannot be well interpreted, as no optimal animal to plant protein (or vice versa) ratio in the diet has been established yet.

Sugar intake was also significantly high for both groups (evaluated against WHO recommended intake) and previous data on HG adults have also shown similarly immoderate amounts of sugar intake [86]. Sugar, coming predominantly from glucose- or fructose-sweetened beverages and confectionary, is a great public health challenge in Hungary [89] and recent analyses have shown two-thirds of the adult population being overweight or obese, which has been partially linked to the excessive consumption of sugar-sweetened beverages [86,90]. Governmental legislative initiatives have been introduced to tackle the situation, with most notably the *2011 Public Health Product Tax Act*, applied to non-staple foods including sugar-sweetened beverages and pre-packaged sweets. However, given our results and the urgency to tackle the current ‘*sugar epidemic*’, measures aimed at reducing excessive sugar consumption should go beyond legal actions and additional regulatory mechanisms should be introduced, particularly targeting early exposures in childhood and adolescence. Such mechanisms may include regulating and monitoring advertising of unhealthy foods and beverages, with special attention to child-directed food marketing.

Moreover, sodium intake was observed to be exceedingly high for HR, both in terms of absolute and adjusted intake, and not significantly different from HG. It should be noted that the estimated intake in this analysis may be subject of a diluted underestimation effect, as sodium is typically under-reported when using recall methods, such as ours [91]. This means that the actual intake may be even higher. Such results strengthen the importance of existing national targets to reduce sodium intake at the population level.

Another critical aspect of diet composition, in both study groups, was the inadequate fiber intake, which has been linked to higher risk of colorectal cancer [92]. This dietary aspect is highly relevant for Hungary as colorectal cancer is one of the most common causes of cancer-related death, for both males and females, in Hungary [93], making it the number one country in the world ranking list of age-standardized mortality (per 100,000) caused by colorectal cancer [94,95]. Recent results from meta-analysis of prospective studies indicate a 10% reduction in colorectal cancer risk for each 10 g/day intake of total dietary fiber and cereal fiber, and an about 20% reduction for each three servings (90 g/day) of whole grain daily, and further reductions with higher intake [96]. Considering the current evidence on the protective effect of dietary fibers, our findings have important public health implications and provide support for public health nutrition recommendations to increase intake of fiber especially in the prevention of colorectal cancer. Fiber intake, combined with potassium estimates, can be an indication of fruits and vegetables consumption, which seems to be low and this is also reflected in the lower micronutrient intake among HR (particularly B-vitamins). Such results are supported by our recent study among HR showing that the Roma participants reported significantly less frequent consumption of fresh fruits and vegetables than the Hungarian subjects [97].

Previous research on Roma attempting to characterize eating habits and food choices during different periods of the year than the current research (April–September), as well as different locations where Roma reside, have also recorded an unfavorable dietary picture. Roma youth from Slovakia were reported to consume fruits less frequently than non-Roma individuals [21]. Roma participants (over 18 years of age) in survey from South Bohemian Region during June 2015 to March 2016, reported high consumption of sugar sweetened beverages and inadequate consumption of fruits and vegetables [24]. Another survey involving Roma and non-Roma subjects, conducted in the latter region, also reported low fruits and vegetables consumption [31]. A report from a European project involving Roma individuals from seven different European countries (i.e., Bulgaria, Czech Republic, Greece, Portugal, Romania, Slovakia, Spain) conducted in varying periods of the year (from March–June and September–December 2008) found that less than a third of Roma participants reported consuming fruits and fresh vegetables on a daily basis, while 36% reported consuming sweets and confectionery every day [26]. Research involving Roma from the county of Rimavská Sobota district of Slovakia (the survey was carried out during June 2007–May 2008) showed that their diet was characterized by low consumption of dairy products and vegetables [27]. A cross-sectional study over a one-year period in 2010, using an instrument with questions related to the frequency of consumption for certain food groups among 400 participants from Roma communities in five districts in Albania, found infrequent consumption of fruits and milk and moderate consumption of meat and vegetables [28]. Furthermore, estimates of the Household Budget Survey in Romania involving almost 9000 settlements during 2004 to 2011, suggest that the Roma population has an inferior diet compared to that of non-Roma populations in terms of a lower proportion of dairy product, fruit, and vegetable intake [29]. Other studies involving Roma participants also suggest excessive consumption of fast-food, fatty meats and sweets and low consumption of vegetables and fruits [25,30]. Eventually, findings from two paired health surveys, that we carried out in the general Hungarian and Roma populations using the same methodology before and after the Decade of Roma Inclusion were compared, and it was clearly shown that the distribution of BMI worsened significantly among younger Roma individuals (in both sexes) between 2004 and 2015, with obesity becoming significantly more frequent [3]. Such findings support the assumption that unhealthy diet is characteristic of the nutritional profile of the Hungarian Roma population, since obesity is a potential consequence of poor dietary behavior [98].

Despite the fact that all the aforementioned studies have not quantified intake of micro- and macro-nutrients among Roma, it appears that dietary quality of Roma is less favorable than that of the host population, regardless of where they live or when the dietary survey was conducted. Our results are in line with these findings. This in turn signifies the need for public health nutrition interventions in addition to existing ones in Hungary, going beyond just legal and regulatory policies [99], while deliberately engaging minorities such as Roma, aiming at modifying current dietary patterns. Given the complexity of nutritional behaviors and the wide range of influences on diet, such efforts require active collaboration of a variety of actors throughout the food system, along with policies targeting multiple sectors. Many populations, as in the present case, are dynamically enriched by a range of ethnic groups and such minority groups should be key targets when tackling inequalities in health. An opportunity exists, particularly within the current framework of the UN Decade of Action on Nutrition (2016–2025) global work program, which can be a successful decade also for Hungary. Our results suggest that such actions should emphasize and reinforce the relevance of more plant-based proteins, higher fiber, fruit, vegetables, whole grains intake and substitution of detrimental fatty acids sources with beneficial fatty acids sources in energy balanced conditions. The present findings results can also imply the presence of dietary risk factors and signify elevated risk for diet-related NCDs in both groups examined. Given the importance of maintaining a healthy diet in supporting health and function, our results are of concern and require further work to be confirmed, as well as to identify the factors that drive such poor dietary patterns and thus, facilitate targeted interventions to Roma, who may need it most.

Eventually, with relevance to the current Corona Virus Disease 2019 (COVID-19) situation, it is important to emphasize that research has showed that major risk factors for hospitalization, severity and mortality of COVID-19 include diet-related conditions, such as obesity, hypertension and type 2 diabetes [100,101,102]. Hence, nutritional well-being for all, particularly the most vulnerable, has heightened significance in the face of the COVID-19 pandemic [103], thus addressing malnutrition in all its forms and diet-related NCDs are crucial in preparedness and building health resilience of populations for this and future public health threats [104]. A streamlined response to COVID-19 in the context of nutrition and NCDs is important to optimize public health outcomes and reduce the impacts of this pandemic on individuals, vulnerable groups, minorities and societies [105].

### Strengths, Limitations and Considerations for Future Research

Our study provides a comprehensive comparative dietary analysis, offering an opportunity to explore diet quality among Roma in relation to a variety of measures of nutritional quality and anthropometric status. However, even though our analysis may be the first comprehensive and detailed characterization for Roma nutrition, there are some limitations to our observations that should be recognized. Observations are based on a double multiple-pass 24-h dietary recall, and even though it is a valid approach to assess dietary intake patterns in epidemiological studies, findings need to be interpreted with caution, as long-term or seasonal variation of dietary patterns, in the populations under investigation, may not be fully captured.

In our study the representation of females among HR was higher than among HG. This has also been the case in our previous surveys conducted among segregated Roma colonies in Hungary, with more female respondents [18], and also in Roma surveys in other countries (we have described in our previous work [44] that one of the major limitations of the study is that females are overrepresented in the Roma sample. This cross-sectional survey was based on randomly selected households and in many households, only women were home during the day when most visits took place, while men had travelled at least locally for public work. The Hungarian government has quadrupled the budget for public works between 2010 and 2015 for all Hungarian municipalities. This is especially relevant for villages in the North-eastern region of Hungary, where segregated Roma settlements are concentrated. Therefore, the majority of workers participating in the program have been men from deprived Roma communities. The same challenge has been identified in a cross-sectional survey among Slovakian Roma living mainly in segregated colonies, where females were strongly overrepresented (64.8%) in the sample [106].

Furthermore, the Roma study population can be considered representative only of HR living in segregated colonies of Northeast Hungary, but not representative of the overall Roma population living in Hungary. A common challenge in ethnicity-based studies is the accurate determination of ethnicity. In the present study, Roma ethnicity status was self-reported, which may result in losing potential participants [8]. Concerning the fact that data collection was made by Roma university students with the support of the local Roma self-government only slight, if any, loss of subjects can be assumed. Considering that more than 8% of the Hungarian population is Roma, it is reasonable to suppose that individuals belonging to the Roma population may be present in the HG sample, which may result in a potential slight underestimation of differences between the two study groups.

Roma who have been, to various degrees, assimilated with the general Hungarian population were not included due to the scope of the study. Additionally, PBF estimates were based on anthropometric equations and future research should consider measuring body composition directly to confirm our results. However, it is remarkable that, regardless of the equation used, there was significantly higher estimated body fatness among HR. Another element that needs to be taken into consideration for the current results is that the NutriComp Étrend ver. 3.03 software does not provide within-person variation corrections of dietary data. To reduce day-to-day variation of dietary intake, the software calculated the average of the two-day intake, allowing room for some measurement error. However, we used this software as it is the only software with nutritional data tailored specifically for the Hungarian context, containing special dishes, and food types consumed only in Hungary and their respective nutrient composition. If the results had been processed with other software, the measurement error would have been unacceptably larger compared with this software.

Eventually, comparisons of dietary patterns and nutritional status between HR and HG, should consider the social context in which such differences are measured and occur, with cautious interpretations that consider the social determinants of health. Research shows that the unemployment rate among HR communities exceeds not only that of the general population, but the unemployment is significantly different even among small areas of Hungary as well [107]. Low levels of education, coupled with widespread discrimination in employment, exclude large numbers from the labor market [108]. Hence, a substantial proportion of the HR population remains economically challenged. This is highly relevant regarding nutrition, as higher-quality diets of lower energy density are likely to cost more [109,110,111,112] and can be not only more costly per kilocalorie, but also more likely to be consumed by individuals with higher educational level as well [109,111]. Therefore, identifying dietary patterns that are nutrient-rich, affordable, and taste-appealing for the HR should be a public health research priority in order to identify and address social inequalities in nutrition and health.

## 5. Conclusions

In summary, the current dietary profile and nutritional status of HR living in segregated colonies in Northeastern Hungary, was found to be suboptimal, with inadequate nutrient composition and anthropometric status estimates, not strongly different than HG population, but occasionally worse among HR. Ethnic-specific differences exist with regard to meeting nutrient-based dietary recommendations, with Roma being less likely to comply, compared to the HG population. To date, this is the first study, to provide detailed and comparable (with the general population) data on nutrient patterns and intake, as well as extensive anthropometric indices in a relatively large sample of Roma. Such data are valuable for developing and implementing public health nutrition strategies to meet dietary recommendations, as well as for guiding nutrition education and intervention programs to reduce the risk of malnutrition in all its forms and diet-related NCDs risk, in this high-risk population. This study also demonstrates the data gaps on intake for key nutrients among HR, highlighting the importance of establishing and integrating Roma nutrition in national surveillance and monitoring systems for key dietary risk factors. It is timely to reconsider dietary guidelines for Hungary, with incorporation of evidence on ethnicity-related considerations. Further research is warranted to elucidate the drivers and possible options for addressing malnutrition in all its forms among HR, as well as HG.

## Figures and Tables

**Figure 1 nutrients-12-02836-f001:**
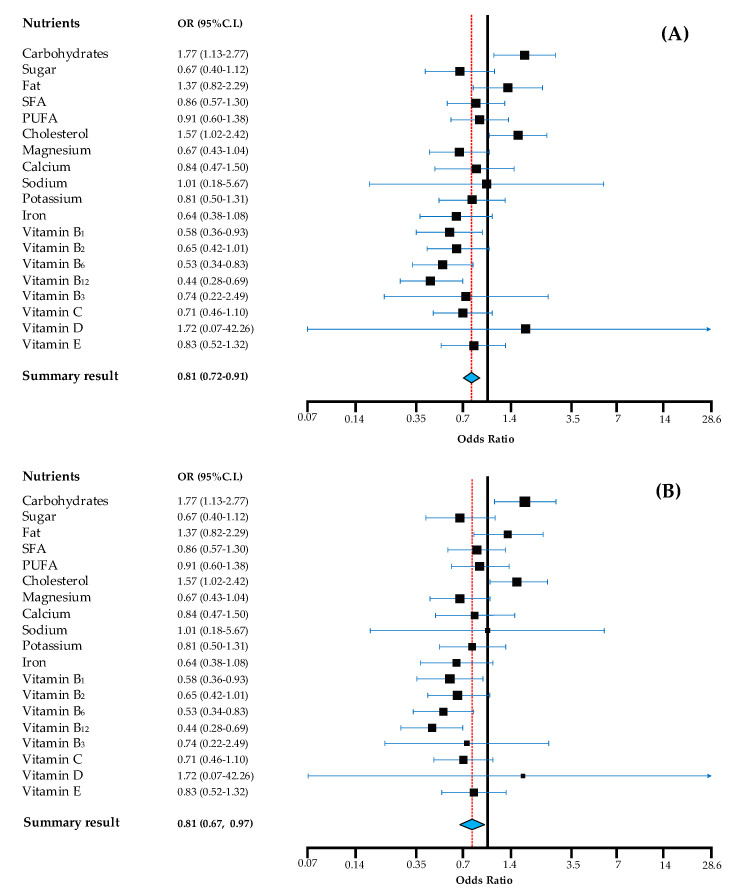
Odds for achieving recommendations of Hungarian Roma compared to Hungarian general population. Note: in the fixed effects model (**A**) the summary result provides the best estimate of an assumed same effect of all nutrients in achieving recommendations and in the random effects model (**B**) the summary result gives the average from the distribution of random effects across nutrients. Calculations are based on age and sex-specific nutrient recommendations according to WHO, where applicable. OR (odds ratio) estimates are adjusted for age, gender, education, marital and perceived financial status, with HG as reference. 95% CI: 95% confidence interval of the mean; PUFA: polyunsaturated fatty acids; SFA: saturated fatty acids.

**Table 1 nutrients-12-02836-t001:** Socio-demographic characteristics of participants.

Variable	Hungarian General (*n* = 359)	Hungarian Roma (*n* = 344)	*p* ^a^
**Age group (*years*)**	44.2 ± 12.2 *	42.9 ± 12.1	>0.05
20–34	93 (25.9%) ^†^	103 (29.9%)	
35–44	92 (25.6%)	85 (24.7%)	
45–54	100 (27.9%)	93 (27.1%)	
55–64	74 (20.6%)	63 (18.3%)	
**Sex (*females*)**	188 (52.4%)	248 (72.1%)	<0.01
**Educational level**			
*Elementary*	76 (21.2%)	292 (84.9%)	<0.01
Secondary	118 (32.9%)	17 (4.9%)	
Vocational training	112 (31.2%)	35 (10.2%)	
University degree	53 (14.7%)	0 (0.0%)	
**Perceived financial status**			
Very good	18 (5.2%)	5 (1.5%)	<0.01
Good	97 (27.6%)	46 (13.5%)	
Fair	190 (54.1%)	186 (54.7%)	
Challenging	40 (11.4%)	85 (25.0%)	
Very challenging	6 (1.7%)	18 (5.3%)	
**Economic activity**			
Full-time employment	267 (74.4%)	233 (67.7%)	<0.01
Part-time employment	29 (8.1%)	23 (6.7%)	
Student	8 (2.2%)	0 (0.0%)	
Retired	22 (6.1%)	22 (6.4%)	
Ill-health retirement	18 (5.0%)	10 (2.9%)	
Unemployed	15 (4.2%)	56 (16.3%)	

^a^ Student *t*-test was used to test differences and chi-square for associations. * Mean ± standard deviation. ^†^ Values are given as number (percentage). Note: For ‘Perceived financial status’ 8 and 4 responses were missing for HG and HR respondents respectively.

**Table 2 nutrients-12-02836-t002:** Anthropometric characteristics of the study populations.

Variable	Hungarian General (*n* = 359)	Hungarian Roma (*n* = 344)	*p* ^a^
**Height (*cm*)-mean (*95% CI*) ^†^**	169.1 (168.1–170.1)	161.4 (160.4–162.4)	<0.01
Males	175.6 (174.4–176.8)	<0.01	<0.01
Females	163.1 (162.1–164.1)	<0.01	<0.01
**Weight (*kg*)-mean (*95% CI*)**	78.0 (76.3–79.8)	72.2 (70.2–74.2)	<0.01
Males	85.0 (82.6–87.3)	81.5 (77.8–85.3)	>0.05
Females	71.8 (69.5–74.0)	68.6 (66.4–70.9)	>0.05
**BMI (*kg/m*^2^)-mean (*95% CI*)**	27.3 (26.7–27.8)	27.7 (26.9–28.4)	>0.05
Males	27.5 (26.8–28.2)	28.0 (26.7–29.3)	>0.05
Females	27.0 (26.1–27.9)	27.5 (26.7–28.4)	>0.05
Underweight	10 (2.8%)	22 (6.4%)	<0.01
Normal weight	116 (32.3%)	109 (31.7%)	
Overweight	129 (35.9%)	84 (24.4%)	
Obese (total)	104 (29.0%)	129 (37.5%)	
Obese class I	73 (20.3%)	83 (24.1%)	>0.05
Obese class II	22 (6.1%)	31 (9.0%)	
Obese class III	9 (2.5%)	15 (4.4%)	
**Metabolically healthy obesity-% (95% CI) ***			
Meigs et al. criteria [61]	33.7 (28.8–38.9)	25.6 (21.1–30.5)	>0.05
Lynch et al. criteria [62]	15.6 (12.0–19.8)	14.8 (11.2–19.0)	>0.05
Karelis et al. criteria [63]	18.9 (15.0–23.4)	18.6 (14.6–23.1)	>0.05
Wildman et al. criteria [64]	21.2 (17.1–25.8)	16.9 (13.1–21.2)	>0.05
**Waist circumference (*cm*) mean (95% CI)**	95.9 (94.4–97.5)	95.1 (93.3–96.9)	>0.05
**WHtR**	♂0.56; ♀0.58	♂0.58; ♀0.60	<0.05
**Abdominal/central obesity IDF_EURO_^1^*–**n (%)***	♂66 (38.6%);	♂42 (43.8%);	<0.05
	♀122 (64.9%)	♀161 (64.9%)	>0.05
**Estimated percentage of body fat-% (95% CI)**			
Gomez-Ambrozi et al. [78]	32.7 (31.7–33.6)	35.1 (34.1–36.2)	<0.01
Deurenberg et al. [57]	30.7 (29.7–31.7)	33.2 (32.1–34.3)	<0.01
Woolcott et al. [55]	22.2 (21.4–23.1)	25.6 (24.6–26.6)	<0.01
Gallagher et al. [58]	29.6 (28.7–30.5)	31.9 (30.9–32.9)	<0.01

^a^ Student *t*-test was used to test differences and chi-square for associations. Significant statistical comparisons are highlighted in grey. * Proportion of subjects that suffer from obesity (BMI ≥ 30 kg/m^2^) but are metabolically healthy according to different published criteria. Values are given as proportion and 95% confidence interval for the proportion. ^†^ Values are given as means (95% confidence interval of the mean). ^1^ International Diabetes Federation, 2006: criteria for Europeans (WC_♂_ ≥ 102 cm and WC_♀_ ≥ 88 cm). WHtR: waist-to-height ratio; PBF: percentage of body fat estimations by different equations; ♂: males; ♀: females. Note: Criteria for determining metabolically healthy obesity were based on different criteria as follows: Meigs et al. criteria (none of the following): (1) Homeostatic Model Assessment for Insulin Resistance (HOMA-IR) ≥75th percentile in study population; Lynch et al. criteria (none of the following): (1) blood pressure (mmHg) >130/85, (2) blood pressure medications, (3) lipid medications, (4) fasting TG/HDL-cholesterol ratio >1.65 in men or >1.32 in women, (4) fasting glucose (mg/dL) >100, and (5) diabetes medications; Karelis et al. criteria (≤1 of the following): (1) fasting TG (mg/dL) ≥150, (2) total cholesterol (mg/dL) ≥200, (3) Low-Density Lipoproteins (LDL)-cholesterol (mg/dL) ≥130, (3) HDL-cholesterol (mg/dL) <50 in men and women, and (4) HOMA-IR >1.9; Wildman et al. criteria (≤1 of the following): (1) blood pressure (mmHg) ≥130/85, (2) blood pressure medications, (3) fasting TG (mg/dL) ≥150, (4) HDL-cholesterol (mg/dL) <40 in men or <50 in women, (5) lipid medications, (6) fasting glucose (mg/dL) ≥100, (7) diabetes medications, (8) HOMA-IR >5.13 (i.e., ≥90th percentile in study population), and (9) C-reactive protein (mg/L) ≥90th percentile in study population. Further, estimations of percentage of body fatness were based on different equations as follows: Gomez-Ambrosi et al. equation: percentage of body fat (PBF) = −44.988 + 0.503 × age + 10.689 × sex + 3.172 × BMI −0.026 × BMI^2^ + 0.181 × BMI × sex −0.02 × BMI × age −0.005 × BMI^2^ × sex + 0.00021×BMI^2^ × age (Sex: Males = 0, Females = 1; R^2^ = 0.79, root mean square error (RMSE) = 4.7%); Deurenberg et al. equation: PBF = −11.4 × sex + 0.20 × age + 1.294 × BMI −8.0 (R^2^ = 0.88, RMSE = 2.5%); Woolcot et al. equation: PBF = 64− (20 × (height/waist)) + (12 × sex) (R^2^ = 0.84, RMSE = 3.5%); and Gallagher et al. equation: PBF = 64.5–848 × (1/BMI) + 0.079 × age −16.4 × sex + 0.05 × sex × age + 39.0 × sex × (1/BMI) (R^2^ = 0.86, RMSE = 4.98%).

**Table 3 nutrients-12-02836-t003:** Total average daily energy intake (*kcal*) by sex and ethnicity.

	Hungarian General (*n* = 359)		Hungarian Roma (*n* = 344)		*p* *
	*Mean*	*95% CI*	*Mean*	*95% CI*	
Both sexes	2188.3	2111.2–2265.3	2114.1	2042.3–2185.8	0.166
Males	2270.9	2148.9–2392.8	2212.5	2064.2–2360.8	0.559
Females	2113.1	2016.5–2209.7	2076.0	1994.5–2157.5	0.561

* Student *t*-test was used to test differences between groups. 95% CI: 95% confidence interval of the mean.

**Table 4 nutrients-12-02836-t004:** Macronutrient intakes among Hungarian Roma and general populations.

Macronutrients	Recommendation [*Ref*.]	Hungarian General (*n* = 359)	Hungarian Roma (*n* = 344)	β (95% CI) †
*Carbohydrates (%E)*	55–75%E [67]	46.2 (45.3;47.1)	48.2 (47.2;49.2)	2.8 (0.9;4.8) *
Sugar (*g*)	**≤ 31 g** [73]	96.27 (89.03;103.5)	101.5 (94.1;108.8)	**3.7 (1.8;5.6) ***
Sugar (*%E*)	**≤10%E (≤5%E)** [67]	17.0 (16.0;18.0)	18.8(17.7;19.8)	**0.03 (0.01;0.05) ***
Fiber (*g*)	**≥24 g** [67]**; ≥42.9 g** [73]	20.3 (19.3;21.3)	20.4 (19.1;21.6)	−2.35 (−4.7;0.01)
Fiber (*g/1000 kcal*)	**14.8 g/1000 kcal** [72]	9.7 (9.2;10.1)	9.9 (9.4;10.4)	−0.75 (−1.7;0.2)
*Proteins (%E)*	**10–15%E** [67]	15.5 (15.2;15.9)	15.1 (14.7;15.4)	−0.59 (−1.3;0.1)
Animal-based proteins (*% tot. proteins*)	**-**	59.3 (57.5;61.0)	60.6 (58.9;62.4)	−1.07 (−2.63;0.49)
Plant-based protein (*% tot. proteins*)	**-**	40.7 (39.0;42.5)	39.4 (37.6;41.1)	0.98 (−0.58;2.54)
Animal/plant protein ratio		1.8 (1.7;1.9)	1.6 (1.5;1.72)	**0.19 (0.04;0.34) ***
Amino acids (*g*)	**-**	76.8 (73.9;79.7)	71.1 (68.5;73.7)	**−2.02 (−3.6;−0.4) ***
Essential amino acids (*g*)	**-**	28.7 (27.6;29.8)	26.4 (25.4;27.4)	**−0.8 (−1.5;−0.2) ***
*Fats*	**15–30%E** [67]	37.1 (36.3;38.0)	36.1 (35.2;37.0)	−1.6 (−3.4;0.2)
Animal-based fats *(% of total fats*)	**-**	59.3 (57.5;61.0)	60.6 (58.9;62.4)	1.69 (−1.77;5.15)
Plant-based fats *(% of total fats*)	**-**	40.7 (39.0;42.5)	39.4 (37.6;41.1)	−3.34 (−6.80;0.12)
SFA (*%E*)	**≤10%E** [67]	10.7 (10.3;11.1)	10.7 (10.3;11.0)	−0.2 (−0.9;0.6)
MUFA (*%E*)	**-**	11.9 (11.5;12.3)	11.4 (11.0;11.8)	−0.5 (−1.4;0.3)
PUFA (*%E*)	**6–10%E** [67]	9.0 (8.7;9.3)	8.2 (7.9;8.5)	**−1.0 (−1.6;−0.4) ***
UFA (*%E*)	**-**	20.9 (20.3;21.4)	19.6 (19.1;20.2)	**−1.5 (−2.6;−0.4) ***
Cholesterol (*mg/1000 kcal*)	**71.4 mg/1000 kcal** [72]	172.9 (164.7;181.0)	159.5 (152.2;166.8)	**−18.8 (−34.4;−3.2) ***
Cholesterol (*mg*)	**<300 mg** [67]**; ≤ 125.2** [73]	369.2 (350.7;387.7)	339.7 (320.3;359.2)	**−41.27 (−80.18;−2.36) ***
ω-3 fatty acids (*%E*)	**1–2%** [67]	0.31 (0.29;0.32)	0.27 (0.26;0.28)	**−0.06 (−0.11;−0.01) ***
ω-6 fatty acids (*%E*)	**5–8%** [67]	8.7 (8.4;9.0)	8.0 (7.7;8.3)	**−0.99 (−1.60;−0.38) ***
α-linolenic acid (*%E*)	**0.5–2%E** [68]	0.27 (0.26;0.28)	0.25 (0.24;0.26)	**−0.03 (−0.05;0.002) ***

* *p* < 0.05; Intake that is significantly different compared to internationally established recommendations are highlighted in grey and significant differences in intake levels between groups are bolded. **^†^** β: regression coefficient (regression coefficient was controlled for the respondent’s age, gender, education, marital and perceived financial status, with Hungarian general population as reference and the β coefficient is associated with HR ethnicity). Notes: every value is given as mean and 95% confidence interval of the mean. 95% CI: 95% confidence interval of the mean; [Ref.]: reference—source of the recommended range; MUFAs: monounsaturated fatty acids; PUFA: polyunsaturated fatty acids; SFA: saturated fatty acids; UFA: unsaturated fatty acids; (%E): intake as percentage of total energy.

**Table 5 nutrients-12-02836-t005:** Micronutrient intakes among Hungarian Roma and general populations.

Micronutrients	Recommendation [Ref.]	Hungarian General (*n* = 359)	Hungarian Roma (*n* = 344)	β (95% CI) ^†^
Minerals and Trace Elements				
Magnesium (*mg/1000 kcal*)	**≥238 mg/1000 kcal** [72]	188.7 (164.7;212.6)	180.0 (172.6;187.3)	−32.2 (−73.1;8.7)
Calcium (*mg/1000 kcal*)	**≥590 mg/1000 kcal** [72]	246.9 (232.5;261.4)	245.9 (233.3;258.4)	1.0 (−27.7;29.7)
Sodium (*mg/1000 kcal*)	**≤1143 mg/1000 kcal** [72]	2605.1 (2508.7;2701.5)	2434.9 (2348.7;2521.2)	**−282.8 (−480.7;−84.9) ***
Sodium (*mg*)	**≤2000 mg** [70]	5644.0 (5351.9;5936.0)	5094.4 (4866.0;5322.8)	**−765.0 (−1304.5;−225.5) ***
Potassium (*mg/1000 kcal*)	**≥2238 mg/1000kcal** [72]	1371.8 (1297.4;1446.1)	1426.8 (1345.8;1507.7)	−105.9 (−267.6;55.8)
Potassium (*mg*)	**≥3510 mg** [71]	2981.8 (2752.2;3211.4)	2971.6 (2778.2;3165.1)	−432.3 (−870.4;5.9)
Iron (*mg/1000 kcal*)	**-**	5.2 (5.0;5.5)	5.2 (4.9;5.5)	**−0.6 (−1.2;−0.1) ***
Iron (*mg*)	**1.05 mg** [69]	11.2 (10.6;11.8)	11.1 (10.2;11.9)	**−1.6 (−3.1;−0.1) ***
*Vitamins*				
Vitamin A (*μg/1000 kcal*)	**-**	140.9 (124.9;156.8)	166.4 (129.1;203.8)	−19.0 (−81.6;43.6)
Vitamin A (*μg RE*)	**500 μg RE** [69]	294.89 (260.9;328.9)	393.1 (279.6;506.69)	−78.86 (−245.64;87.92)
Beta-carotene (*mg/1000 kcal*)	**-**	1.2 (1.1;1.3)	1.36 (1.19;1.53)	−0.13 (−0.44;0.17)
Vitamin B_1_ (*μg/1000 kcal*)	**-**	465.4 (448.8;482.1)	457.1 (439.2;474.9)	−32.9 (−68.4;2.5)
Vitamin B_1_ (*μg*)	**≥1100 *μ*g** [69]	1023.1 (973.3;1073.0)	960.5 (912.3;1008.8)	**−109.9 (−207.9;−11.9) ***
Vitamin B_2_ (*μg/1000 kcal*)	**-**	567.7 (512.4;622.9)	539.1 (514.0;564.1)	−68.51 (−164.6;27.6)
Vitamin B_2_ (*μg*)	**≥1100 *μ*g** [69]	1290.3 (1041.7;1538.9)	1135.4 (1068.1;1202.7)	−355.36 (−728.62;17.89)
Vitamin B_6_ (*μg/1000 kcal*)	**-**	813.2 (785.2;841.2)	771.8 (741.7;802.0)	**−103.8 (−162.4;−45.3) ***
Vitamin B_6_ (*μg*)	**≥1300 μg** [69]	1761.7 (1689.0;1834.4)	1591.8 (1518.4;1665.1)	**−270.7 (−415.9;−125.6) ***
Vitamin B_12_ (*μg/1000 kcal*)	**-**	1.6 (0.7;2.5)	1.3 (0.9;1.8)	**−2.3 (−3.9;−0.6) ***
Vitamin B_12_ (*μg*)	**≥2.4 μg** [69]	3.7 (1.4;6.1)	3.0 (1.8;4.1)	−5.58 (−9.30;−1.87) *
Vitamin B_3_ (*mg NE/1000 kcal*)	**≥6.6 mg NE/1000 kcal** [74]	9.7 (8.1;11.3)	8.4 (8.0;8.8)	**−3.7 (−6.4;−1.0) ***
Vitamin B_3_ (*mg NE*)	**≥14 mg NE** [69]	22.9 (15.7;30.2)	17.7 (16.6;18.7)	**−12.8 (−23.3;−2.3) ***
Vitamin C (*mg/1000 kcal*)	**-**	37.3 (33.9;40.7)	40.08 (35.56;44.6)	**−8.7 (−16.7;−0.6) ***
Vitamin C (*mg*)	**≥45 mg** [69]	78.8 (71.7;86.0)	79.4 (71.3;87.5)	**−18.98 (−34.26;−3.7) ***
Vitamin D (*mg/1000 kcal*)	**-**	0.8 (0.7;0.9)	0.8 (0.6;0.9)	−0.1 (−0.3;0.2)
Vitamin D (*μg*)	**≥10 μg** [69]	1.7 (1.5;1.9)	1.7 (1.4;2.0)	−0.23 (−0.72;0.27)

* *p* < 0.05; Intake that is significantly different compared to internationally established recommendations are highlighted in grey and significant differences in intake levels between groups are bolded. **^†^** β: regression coefficient (regression coefficient was controlled for the respondent’s age, gender, education, marital and perceived financial status, with Hungarian general population as reference and the β coefficient is associated with HR ethnicity). Notes: Every value is given as mean and 95% confidence interval of the mean. 95% CI: 95% confidence interval of the mean; [Ref.]: reference—source of the recommended range; (%E): intake and as percentage of total energy; NE: niacin equivalents; RE: retinol equivalents.

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
