# Peer review of "Dietary Profile and Nutritional Status of the Roma Population Living in Segregated Colonies in Northeast Hungary"

_nutrients, 2020, doi:10.3390/nu12092836_

Round 1
Reviewer 1 Report
Introduction
In my opinion, the introduction section lacks of a clear focus on the background specific to the research questions. It does not become clear why it is important to investigate “dietary profile and nutritional status of the Roma population living in segregated colonies in Northeast Hungary”.
In addition, the introduction needs to be improved to concentrate on the study aim and hypotheses.
Materials and Methods
This section is incomplete and should be supplemented with information regarding the inclusion/exclusion criteria in the study (how were participants selected?)
Results
- I observed an inaccuracy in Table 1. The percent for some variables were incorrect based on their sample size: educational level (the sum is 100,1% in HG), perceived financial status (the sum is 99,9% in HG, n = 351 vs n = 359) and the other variables in the table. Please do a check and recalculate.
- In Table 2 there is a lack of information about Meigs et al. criteria, Lynch et al. criteria, Karelis et al. criteria and Wildman et al. criteria as well as about estimated percentage of body fat proposed by Gomez-Ambrozi et al., Deurenberg et al., Woolcott et al. and Gallagher et al. These information should be completed.
- There is a lack of R-Squared value and ΔR2.
Discussion
The discussion needs to go more in depth in order to develop a better understanding of the topic. A more detailed outlook on the continuation of the study considering the results should be presented.
Limitation of the study
Some other weaknesses or limitations of this study should be also provided (e.g. line 91, p.2: “Ethnicity of the colony population was assessed by self-declaration”)
Author Response
Introduction
In my opinion, the introduction section lacks of a clear focus on the background specific to the research questions. It does not become clear why it is important to investigate “dietary profile and nutritional status of the Roma population living in segregated colonies in Northeast Hungary”. In addition, the introduction needs to be improved to concentrate on the study aim and hypotheses.
We have introduced some modifications to the introduction in order to address Reviewer’s remarks, as well as to enhance readability and clarity of this section, as follows:
- Lines 47-51: we deleted this part;
- Line 61: added ‘In addition’ in order to improve readability and transition from one paragraph to the other;
- Lines 67-69: ‘Apart from a recent attempt to qualify dietary intake of Roma [31], nutrient intake patterns and data among Roma are not available yet in Hungary or elsewhere.’
- Lines 73-74: We have deleted this part
- Lines 78-79: We added the following statement for more context: ‘The spatial patterns of Roma show their intense segregation and peripheralization mainly in Northeast Hungary and South Transdanubia [42] ‘
- Lines 81-86: We have added more context and information for the study as follows: ‘Since these new nutrition-related challenges are arising among HR, it is necessary to have a clear understanding of the dietary profile and nutritional status of this underserved ethnic group, to better address malnutrition challenges in all their forms, as well as commit to the Rome Declaration on Nutrition’s common vision to eradicate all forms of malnutrition, in the framework of UN Decade of Action on Nutrition (2015-2025).’
- Line 89: We have added ‘and nutrient’ to let the reader know the primary focus of the dietary analysis
The aim of our study and our hypothesis has also been clarified as the Reviewer recommended, as follows (lines 90-99):
‘We hypothesized that the more unfavorable health status of Roma in comparison with the general population is partly related to their unhealthier dietary profile and nutritional status. Hence, examining dietary profile and nutritional status characteristics of Roma, particularly those living in highly deprived segregated settlements, can provide an enhanced understanding on their nutrition and its potential interrelationship (s) with their health, which in turn can help to inform and guide policies and decision-making. To test our hypothesis, we used data obtained from our Complex Health (Interview and Examination) Survey [42], with the intention of investigating nutritional characteristics and dietary behavior of this ethnic group living in segregated colonies in Northeast Hungary – where they primarily concentrated.’
Materials and Methods
This section is incomplete and should be supplemented with information regarding the inclusion/exclusion criteria in the study (how were participants selected?)
Results
I observed an inaccuracy in Table 1. The percent for some variables were incorrect based on their sample size: educational level (the sum is 100,1% in HG), perceived financial status (the sum is 99,9% in HG, n = 351 vs n = 359) and the other variables in the table. Please do a check and recalculate.
‘Note: For ‘Perceived financial status’ 8 and 4 responses were missing for HG and HR respondents respectively.’
In Table 2 there is a lack of information about Meigs et al. criteria, Lynch et al. criteria, Karelis et al. criteria and Wildman et al. criteria as well as about estimated percentage of body fat proposed by Gomez-Ambrozi et al., Deurenberg et al., Woolcott et al. and Gallagher et al. These information should be completed.
There is a lack of R-Squared value and ΔR2.
In order to address Reviewer’s comment on the lack of explanation for the estimation of body fatness, as well as metabolically healthy obesity estimates, we have added an elaboration in the footnotes of Table 2 in the form of a note as follows (lines 259-276):
‘Note: Criteria for determining metabolically healthy obesity were based on different criteria as follows: Meigs et al. criteria (none of the following): (1) HOMA-IR≥75th percentile in study population; Lynch et al. criteria (none of the following): (1) blood pressure (mmHg)>130/85, (2) blood pressure medications, (3) lipid medications, (4) fasting TG/HDL-cholesterol ratio >1.65 in men or >1.32 in women, (4) fasting glucose (mg/dL)>100, and (5) diabetes medications; Karelis et al. criteria (≤1 of the following): (1) fasting TG (mg/dL) ≥150, (2) total cholesterol (mg/dL) ≥200, (3) LDL-cholesterol (mg/dL) ≥130, (3) HDL-cholesterol (mg/dL)<50 in men and women, and (4) HOMA-IR>1.9; Wildman et al. criteria (≤1 of the following): (1) blood pressure (mmHg) ≥130/85, (2) blood pressure medications, (3) fasting TG (mg/dL) ≥150, (4) HDL-cholesterol (mg/dL) <40 in men or <50 in women, (5) lipid medications, (6) fasting glucose (mg/dL) ≥100, (7) diabetes medications, (8) HOMA-IR>5.13 (i.e., ≥90th percentile in study population), and (9) C-reactive protein (mg/L) ≥90th percentile in study population. Further, estimations of percentage of body fatness were based on different equations as follows: Gomez-Ambrosi et al. equation: PBF = - 44.988 + 0.503 × age + 10.689 × sex + 3.172 × BMI - 0.026 × BMI2 + 0.181 × BMI × sex - 0.02 × BMI × age - 0.005 × BMI2 × sex + 0.00021×BMI2×age (Sex: Males = 0, Females = 1; R2 = 0.79, Root Mean Square Error (RMSE) = 4.7%); Deurenberg et al. equation: PBF = - 11.4 × sex + 0.20 × age + 1.294 × BMI - 8.0 (R2 = 0.88, RMSE = 2.5%); Woolcot et al. equation: PBF = 64 − (20 × (height/waist)) + (12 × sex) (R2 = 0.84, RMSE = 3.5%); and Gallagher et al. equation: PBF = 64.5 – 848 × (1 / BMI) + 0.079 × age - 16.4 × sex + 0.05 × sex × age + 39.0 × sex × (1/BMI) (R2 = 0.86, RMSE = 4.98 %)’
We have added the R2 values of equations that we used for calculating body fatness (lines 271-276), but for the rest of the results we do not consider it relevant to insert additional R2 and ΔR2 values, based on the nature of our regression analyses which was performed with the aim of adjusting and correcting comparison of nutrients, rather than explaining the variation that is captured by our regression models. The latter may be relevant, in our view, only for the validation of equations for the estimated percentage of body fatness (PBF) that we have already indicated next to each equation in the footnotes of Table 2.
Discussion
The discussion needs to go more in depth in order to develop a better understanding of the topic. A more detailed outlook on the continuation of the study considering the results should be presented.
We have revised the Discussion section to address Reviewer’s remarks. More specifically, we have deleted lines 425-433 and added more in-depth analyses (lines 352-367), as follows:
‘Recently race- and ethnicity-based health disparities have become a central focus of public health research, practice, and policy, as a growing body of evidence shows a strong association between racial/ethnic and socio-economic disparities with healthy dietary and nutrient patterns [80-82]. Diet-related disparities “ play an important role and exist in the form of “differences in dietary intake, dietary behaviors, and dietary patterns in different segments of the population, resulting in poorer dietary quality and inferior health outcomes for certain groups and an unequal burden in terms of disease incidence, morbidity, mortality, survival, and quality of life.” [83]. Since diet and nutrition are closely related to a number of noncommunicable diseases, there is growing interest in characterizing the association between dietary and nutrient intake in specific disadvantaged minority population groups, such a Roma. In Europe, Roma population constitutes the largest ethnic minority (estimated to be between 10-12 million) [1] and has been a major focus of ethnicity-based studies in past decades [2,3]. The poor living conditions in which some Roma people live, frequently on the outskirts of towns and villages and in substandard settlements, allow relatively straightforward identification of locations in which Roma people are concentrated [46]. This study has taken advantage of this opportunity, by sampling HR participants in Northeastern Hungary, where the Roma population is greatest and in identified settlements, in which the population was almost exclusively Roma.’
We have also introduced a new elaboration of the results and contextualize them with the current known literature (lines 466-502), as follows:
‘Such results are supported by our recent study among HR showing that the Roma participants reported significantly less frequent consumption of fresh fruits and vegetables than the Hungarian subjects [99]. Previous research on Roma attempting to characterize eating habits and food choices during different periods of the year than the current research (April - September), as well as different locations where Roma reside, have also recorded an unfavorable dietary picture. Roma youth from Slovakia were reported to consume fruits less frequently than non-Roma individuals [21] Roma participants (over 18 years of age) in survey from South Bohemian Region during June 2015 to March 2016, reported high consumption of sugar sweetened beverages and inadequate consumption of fruits and vegetables [24]. Another survey involving Roma and non-Roma subjects, conducted in the latter region, also reported low fruits and vegetables consumption [31]. A report from a European project involving Roma individuals from seven different European countries (i.e. Bulgaria, Czech Republic, Greece, Portugal, Romania, Slovakia, Spain) conducted in varying periods of the year (from March – June and September – December 2008) found that less than a third of Roma participants reported consuming fruits and fresh vegetables on a daily basis, while 36% reported consuming sweets and confectionery every day [26]. Research involving Roma from the county of Rimavská Sobota district of Slovakia (the survey was carried out during June 2007 – May 2008) showed that their diet was characterized by low consumption of dairy products and vegetables [27]. A cross-sectional study over a one-year period in 2010, using an instrument with questions related to the frequency of consumption for certain food groups among 400 Roma participants from Roma communities in five districts in Albania, found infrequent consumption of fruits and milk and moderate consumption of meat and vegetables [28]. Further, estimates of the Household Budget Survey in Romania involving almost 9,000 settlements during 2004 to 2011, suggest that the Roma population has an inferior diet compared to that of non-Roma populations in terms of a lower proportion of dairy product, fruit, and vegetable intake [29]. Other studies involving Roma participants also suggest excessive consumption of fast-food, fatty meats and sweets and low consumption of vegetables and fruits [30] [25]. Eventually, findings from two paired health surveys, that we carried out in the general Hungarian and Roma populations using the same methodology before and after the Decade of Roma Inclusion were compared, and it was clearly shown that the distribution of BMI worsened significantly among younger Roma individuals (in both sexes) between 2004 and 2015, with obesity becoming significantly more frequent [3]. Such findings support the assumption that unhealthy diet is characteristic of the nutritional profile of the Hungarian Roma population, since obesity is a potential consequence of poor dietary behavior [100]. Despite the fact that all the above-mentioned studies have not quantified intake of micro- and macro-nutrients among Roma, it appears that dietary quality of Roma is less favorable than that of the host population, regardless of where their live or when the dietary survey was conducted. Our results are in line with these findings.’
Limitation of the study
Some other weaknesses or limitations of this study should be also provided (e.g. line 91, p.2: “Ethnicity of the colony population was assessed by self-declaration”)
To address this limitation, we have added the following statement in Section 4.1 (lines 554-560):
‘A common challenge in ethnicity-based studies is the accurate determination of ethnicity. In the present study, Roma ethnicity status was self-reported, which may result in losing potential participants. Concerning the fact that data collection was made by Roma university students with the support of the local Roma self-government only slight – if any – loss of subjects can be assumed. Considering that more than 8% of the Hungarian population is Roma, it is reasonable to suppose that individuals belonging to the Roma population may be present in the HG sample, which may result in a potential slight underestimation of differences between the two study groups.’

Reviewer 2 Report
Dear authors, your manuscript is very interesting giving some very useful insight in the dietary profile and nutritional status of the Roma population.
Introduction section is clear and supported with background information but need to be revised.
M&M section organized well and explained well. Also, result and discussion section well organized.
Conclusion section is not supported with result and discussion. Conclusion section need to be revised.
Author Response
Dear authors, your manuscript is very interesting giving some very useful insight in the dietary profile and nutritional status of the Roma population.
Introduction section is clear and supported with background information but need to be revised.
M&M section organized well and explained well. Also, result and discussion section well organized.
Conclusion section is not supported with result and discussion. Conclusion section need to be revised.
In the ‘Conclusions’’, we have addressed Reviewer’s suggestion to revise this section. The concluding part in its rewritten form is the following (lines 588-605):
‘In summary, current dietary profile and nutritional status of HR living in segregated colonies in Northeastern Hungary, was found to be suboptimal, with inadequate nutrient composition and anthropometric status estimates, not strongly different than HG population, but occasionally worse among HR. Ethnic-specific differences exist with regard to meeting nutrient-based dietary recommendations, with Roma being less likely to comply, compared to HG population. To date, this is the first study, to provide detailed and comparable (with the general population) data on nutrient patterns and intake, as well as extensive anthropometric indices in a relatively large sample of Roma. Such data are valuable for developing and implementing public health nutrition strategies to meet dietary recommendations, as well as for guiding nutrition education and intervention programs to reduce the risk of malnutrition in all its forms and diet-related NCDs risk, in this high-risk population. This study also demonstrates the data gaps on intake for key nutrients among HR, highlighting the importance of establishing and integrating Roma nutrition in national surveillance and monitoring systems for key dietary risk factors. It is timely to reconsider dietary guidelines for Hungary, with incorporation of evidence on ethnicity-related considerations. Further research in warranted to elucidate the drivers and possible options for addressing malnutrition in all its forms among HR, as well as HG.’

Reviewer 3 Report
Thank you for this interesting and important paper on the diet and nutrition status of the Hungarian Roma population which you have mentioned has not been subject to extensive prior research. The methods are appropriate and the paper is well written and well referenced. Some aspects were not totally clear and would benefit from revisions.
(1) The Introduction references extensive studies that describe dietary risk factors and deleterious diet patterns among Roma communities (references 24-43) but you also have written that “Quantified dietary data among Roma are not available yet” and “dietary intake patterns of HR adults (nor Roma adults in other countries) has never been quantified”. Some of the references you pointed to appear to involve some kinds of quantitative approaches toward diet. Please clarify the aspects of the Roma diet that have not been quantified…intakes of foods, intakes of nutrients, other aspects, etc., and does this lack of quantification apply to Roma generally or just the ones in Hungary? It is helpful to give context in the Introduction and/or Discussion about existing information on intakes of food groups and diet patterns in particular, because you have advised food-based approaches for improving diet among HR and in Hungary in the Discussion, but you have not actually conducted food-based analyses in your study.
(2) The structure of the Introduction section is confusing. Paragraph 1 introduces the Roma community and their disadvantaged health status relative to other Europeans, and some concepts are repeated in paragraph 3. Paragraph 2 introduces dietary risk factors and deleterious diet patterns among Roma and data gaps, and some concepts are repeated in paragraph 4. Please restructure the Introduction so that there is a logical flow, e.g. from background and existing knowledge, knowledge gaps, to the study objective, or something along those lines, without repeating ideas and going backwards.
(3) Section 2.1.2: In this section I would suggest referencing your 2018 study in which you developed and validated the 24 hour recall protocol (I think you referenced it in the Supplementary Material only). Were the weekday and weekend day consecutive? Were two recalls administered to all participants or just a subset? If just a subset, how large was the subset?
(4) Section 2.2: “Dietary data were processed with NutriComp Étrend”. Please explain what data processing is done in NutriComp…just calculating nutrient intakes, or anything else? Does the software correct 24 hour recall measurements for within person variation, and if so, what is the method based on? It is important to correct for this especially if you are assigning measures of spread in dietary intakes and characterizing intakes in reference to recommended levels. If no correction was made, this would seem to be an important limitation that should be noted in the discussion.
(5) Section 2.2: “binary outcomes were created based on international recommendations”. Please be more specific, e.g. that you are referring to international individual nutrient intake recommendations.
(6) Section 3: A seemingly large proportion (12%) of participants were excluded based on implausible energy intakes. Could you please comment on this. Also, it would be helpful to know what percentage was excluded due to implausibly high vs. implausibly low intakes, and to provide some information on socio-demographic characteristics of these participants for comparison with those participants who were included.
(7) Table 1: The Roma group has a substantially higher proportion of females than the general group (72.1% vs. 52.4%). Is this reflective of actual differences in the sex distributions of the target communities (e.g. due to internal migration patterns in Hungary) or is this due to differences in response rate among HR men and women or another sampling issue (based on your knowledge of the survey or other comparison surveys that you know of)? Please address this in the paper. If this is a sampling issue then this would seem to be an important limitation, and may have some implications for your some of your results (discussed below) despite the fact that you controlled for gender in your regression models.
(8) Table 2: It is interesting to be able to compare metabolically healthy obesity and percentage of body fat presented in terms of different reference standards. Thank you for providing those details.
(9) Section 3.2 (and Tables 3-5 and Figure 1): Are the dietary intake statistics based on estimated usual intakes (correcting for within person variation) or are they based on within-person means of the two recall days? This can be clarified Section 2.1.2.
(10) Tables 4 and 5: There are a lot of interesting details in these tables, thanks. It may be helpful if you can clearly note for the reader in the table or the footnote that the beta coefficient is associated with HR ethnicity.
(11) Tables 4 and 5: The referenced nutrient recommendations do not always seem to apply to all age or sex groups. For example it looks like the WHO recommended intakes for vitamin A in women aged 19-65 is listed as 500 ug RE in reference #75 but you have only referenced 600 ug RE (the recommended intake for men, and women 65+). When there are multiple reference values that apply to your study population (i.e. men and women aged 20 to 64 years) it would be helpful to list the different values. In the Hungarian General and Hungarian Roma columns it would also be helpful to give male and female mean intakes for each nutrient, in the main text or in a supplement (even if a nutrient has the same requirement among males and females, it seems awkward to compare intakes between the general and HR populations because the proportion of females in the HR group is much higher), and it would also be interesting to know if there were important male – female differences (statistical tests not needed).
(12) Figure 1: When you assigned the outcome variable (nutrient adequacy) to individuals, did you only use the reference values listed in Tables 4 and 5, or did you use age and sex-specific values (when they were different across groups)? Even though you adjusted for gender, the analysis will not be correct unless you have accurately assigned adequacy at the individual level.
(13) Tables 4 and 5, and Figure 1: When estimating the beta coefficients in tables 4 and 5, and the odds ratios in Figure 1, did you consider or examine potential interactions between ethnicity and other variables that you controlled for (age, gender, education, marriage, financial status)?
(14) The discussion is well written. In Section 4.1 Paragraph 1: Can you please provide some general information from other studies or that you otherwise know of regarding diet during months not covered by your survey (September to April), and between northeast HR and other assimilated HR (this would be helpful to help contextualize the limitations that you have mentioned)?
Author Response
Thank you for this interesting and important paper on the diet and nutrition status of the Hungarian Roma population which you have mentioned has not been subject to extensive prior research. The methods are appropriate and the paper is well written and well referenced. Some aspects were not totally clear and would benefit from revisions.
(1) The Introduction references extensive studies that describe dietary risk factors and deleterious diet patterns among Roma communities (references 24-43) but you also have written that “Quantified dietary data among Roma are not available yet” and “dietary intake patterns of HR adults (nor Roma adults in other countries) has never been quantified”. Some of the references you pointed to appear to involve some kinds of quantitative approaches toward diet. Please clarify the aspects of the Roma diet that have not been quantified…intakes of foods, intakes of nutrients, other aspects, etc., and does this lack of quantification apply to Roma generally or just the ones in Hungary? It is helpful to give context in the Introduction and/or Discussion about existing information on intakes of food groups and diet patterns in particular, because you have advised food-based approaches for improving diet among HR and in Hungary in the Discussion, but you have not actually conducted food-based analyses in your study.
We thank the Reviewer for the thoughtful recommendations regarding the efforts of some studies to analyze dietary data and we have clarified that the dietary characteristics of Roma have not been quantified before this study, by introducing the following modifications (in Track Changes lines 67-69 and lines 89-93):
‘Apart from a recent attempt to qualify dietary intake of Roma [31], nutrient intake patterns and data among Roma are not available yet in Hungary or elsewhere.’
and
‘However, dietary intake and nutrient patterns of HR adults (nor Roma adults in other countries) have never been quantified.’
In addition, we have re-considered the statements about food-based approaches and is now explained in details with mentioning findings of other studies in the Discussion section as follows (lines 469-502):
‘Previous research on Roma attempting to characterize eating habits and food choices during different periods of the year than the current research (April - September), as well as different locations where Roma reside, have also recorded an unfavorable dietary picture. Roma youth from Slovakia were reported to consume fruits less frequently than non-Roma individuals [21] Roma participants (over 18 years of age) in survey from South Bohemian Region during June 2015 to March 2016, reported high consumption of sugar sweetened beverages and inadequate consumption of fruits and vegetables [24]. Another survey involving Roma and non-Roma subjects, conducted in the latter region, also reported low fruits and vegetables consumption [31]. A report from a European project involving Roma individuals from seven different European countries (i.e. Bulgaria, Czech Republic, Greece, Portugal, Romania, Slovakia, Spain) conducted in varying periods of the year (from March – June and September – December 2008) found that less than a third of Roma participants reported consuming fruits and fresh vegetables on a daily basis, while 36% reported consuming sweets and confectionery every day [26]. Research involving Roma from the county of Rimavská Sobota district of Slovakia (the survey was carried out during June 2007 – May 2008) showed that their diet was characterized by low consumption of dairy products and vegetables [27]. A cross-sectional study over a one-year period in 2010, using an instrument with questions related to the frequency of consumption for certain food groups among 400 participants from Roma communities in five districts in Albania, found infrequent consumption of fruits and milk and moderate consumption of meat and vegetables [28]. Further, estimates of the Household Budget Survey in Romania involving almost 9,000 settlements during 2004 to 2011, suggest that the Roma population has an inferior diet compared to that of non-Roma populations in terms of a lower proportion of dairy product, fruit, and vegetable intake [29]. Other studies involving Roma participants also suggest excessive consumption of fast-food, fatty meats and sweets and low consumption of vegetables and fruits [30] [25]. Eventually, findings from two paired health surveys, that we carried out in the general Hungarian and Roma populations using the same methodology before and after the Decade of Roma Inclusion were compared, and it was clearly shown that the distribution of BMI worsened significantly among younger Roma individuals (in both sexes) between 2004 and 2015, with obesity becoming significantly more frequent [3]. Such findings support the assumption that unhealthy diet is characteristic of the nutritional profile of the Hungarian Roma population, since obesity is a potential consequence of poor dietary behavior [100]. Despite the fact that all the above-mentioned studies have not quantified intake of micro- and macro-nutrients among Roma, it appears that dietary quality of Roma is less favorable than that of the host population, regardless of where they live or when the dietary survey was conducted. Our results are in line with these findings.’
(2) The structure of the Introduction section is confusing. Paragraph 1 introduces the Roma community and their disadvantaged health status relative to other Europeans, and some concepts are repeated in paragraph 3. Paragraph 2 introduces dietary risk factors and deleterious diet patterns among Roma and data gaps, and some concepts are repeated in paragraph 4. Please restructure the Introduction so that there is a logical flow, e.g. from background and existing knowledge, knowledge gaps, to the study objective, or something along those lines, without repeating ideas and going backwards.
In order to address the Reviewer’s suggestions and increase readability and coherence of the Introduction we have modified and restructured it as the Reviewer suggests so some concepts and ideas are not repeated.
The Introduction is structured now from background and existing knowledge, knowledge gaps, to the study objective (in Track Changes: lines 47-51, Line 61, lines 67-69, lines 78-79, lines 81-86, line 89, line 91 and lines 92-98). Please see the restructured Introduction in the revised version of the manuscript.
(3) Section 2.1.2: In this section I would suggest referencing your 2018 study in which you developed and validated the 24 hour recall protocol (I think you referenced it in the Supplementary Material only). Were the weekday and weekend day consecutive? Were two recalls administered to all participants or just a subset? If just a subset, how large was the subset?
Thank you for your recommendation to include our previous research. We have included this research as suggested and explained that the weekend day and weekday were not consecutive and that we excluded Monday, since on Monday the food prepared on Sunday is frequently consumed and this may bias estimations. We have added the following statement to clarify this for the readers (lines 151-153):
‘Dietary intake data were obtained in case of each participant through a double (i.e. one non-consecutive weekday [excluding Monday] and one weekend day) interviewer-assisted multiple-pass 24-hour dietary recall protocol developed and validated in our previous study [48].’
Also, we have clarified your comment on whether the two recalls were administered to all survey participants, by adding a statement in the Methods, sub-section 2.1.4. (lines 168-169), as follows:
‘The two dietary recalls were administered to all survey participants.’
(4) Section 2.2: “Dietary data were processed with NutriComp Étrend”. Please explain what data processing is done in NutriComp…just calculating nutrient intakes, or anything else? Does the software correct 24 hour recall measurements for within person variation, and if so, what is the method based on? It is important to correct for this especially if you are assigning measures of spread in dietary intakes and characterizing intakes in reference to recommended levels. If no correction was made, this would seem to be an important limitation that should be noted in the discussion.
First, we have indicated in a statement added to the Data analysis section as follows (lines 188-190):
‘The software converts inputs of food and drinks intakes, into quantified macro- and micro-nutrient intakes.’
The Reviewer raises a very important point when she/he points out the importance of within-person variation correction, which in the case of our software was not possible. However, the database still has important strengths, namely (1) the relatively large sample size conducted among Roma living in segregated colonies, (2) it takes into account the Hungarian specific dietary and culinary context, with recipes and dishes consumed only by this population and (3) we have measured two recalls with average intake values (which does not fully diminish the error potentially caused by the within person variation, but slightly reduces accuracy). Thus, we have added in the limitations section of the Discussion a statement as follows (lines 565-573):
‘Another element that needs to be taken into consideration for the current results is that the NutriComp Étrend ver. 3.03 software does not provide within-person variation corrections of dietary data. In order, to reduce day to day variation of dietary intake, the software calculated the average of the two-day intake, allowing room for some measurement error. However, we used this software as it is the only software with nutritional data tailored specifically for the Hungarian context, containing special dishes, and food types consumed only in Hungary and their respective nutrient composition. If the results had been processed with another software, the measurement error would have been unacceptably larger compared with this software’.
(5) Section 2.2: “binary outcomes were created based on international recommendations”. Please be more specific, e.g. that you are referring to international individual nutrient intake recommendations.
(6) Section 3: A seemingly large proportion (12%) of participants were excluded based on implausible energy intakes. Could you please comment on this. Also, it would be helpful to know what percentage was excluded due to implausibly high vs. implausibly low intakes, and to provide some information on socio-demographic characteristics of these participants for comparison with those participants who were included.
To address this, we have added some notes in the Results section, Paragraph 1 (in Track Changes: lines 232-234) as follows:
‘…were excluded, i.e. (51 HG subjects (4 and 47 for implausibly low intake and high intake, respectively) and 43 HR subjects (4 and 39 for implausibly low intake and high intake, respectively), thus 703 participants aged 20 to 64 years (i.e. 359 HG and 344 HR) were included in the final analysis.’
We have also added two new supplementary Tables S1-S2, which provide more resolution on the excluded implausibly low and high intakes – separately and with characteristics of included participants to allow comparisons for the interested reader. We also added a statement (line 235) to orient the reader to explore the new Tables S1-S2:
‘For more details on the characteristics of excluded subjects see Tables S1-S2.’
(7) Table 1: The Roma group has a substantially higher proportion of females than the general group (72.1% vs. 52.4%). Is this reflective of actual differences in the sex distributions of the target communities (e.g. due to internal migration patterns in Hungary) or is this due to differences in response rate among HR men and women or another sampling issue (based on your knowledge of the survey or other comparison surveys that you know of)? Please address this in the paper. If this is a sampling issue then this would seem to be an important limitation, and may have some implications for your some of your results (discussed below) despite the fact that you controlled for gender in your regression models.
In our study the representation of females among HR was higher than among HG, and such differences can be seen between Roma population living in segregated colonies and the general population and we believe that the higher representation of females among HR can be explained with this fact rather than the response rate. This has also been the case in our previous surveys conducted among segregated Roma colonies in Hungary, with more female respondents [1]. In a recent Slovakian cross-sectional study involving Roma living mainly in segregated colonies, females were also overrepresented in the Roma sample [2]. In order to bring this in the attention of the interested reader we added a statement in the Discussion, under sub-section ‘Strengths, limitations and considerations for future research’ (lines 539-551), as follows:
‘In our study the representation of females among HR was higher than among HG. This has also been the case in our previous surveys conducted among segregated Roma colonies in Hungary, with more female respondents [18], and also in Roma surveys in other countries ( We have described in our previous work [45] that one of the major limitations of the study is that females are overrepresented in the Roma sample. This cross-sectional survey was based on randomly selected households and in many households, only women were home during the day when most visits took place, while men had travelled at least locally for public work. The Hungarian government has quadrupled the budget for public works between 2010 and 2015 for all Hungarian municipalities. This is especially relevant for villages in the North-eastern region of Hungary, where segregated Roma settlements are concentrated. Therefore, the majority of workers participating in the program have been men from deprived Roma communities. The same challenge has been identified in a cross-sectional survey among Slovakian Roma living mainly in segregated colonies, where females were strongly overrepresented (64.8%) in the sample [107].’
(8) Table 2: It is interesting to be able to compare metabolically healthy obesity and percentage of body fat presented in terms of different reference standards. Thank you for providing those details.
We thank the Reviewer for the comment on the value of such indices presented for the first time for the Roma population.
(9) Section 3.2 (and Tables 3-5 and Figure 1): Are the dietary intake statistics based on estimated usual intakes (correcting for within person variation) or are they based on within-person means of the two recall days? This can be clarified Section 2.1.2.
As clarified above the statistics in Tables 3-5 and Figure 1 are based on within-person means of the two recall days. However, in our regression analyses we have made Box–Cox transformation and the use of HG sample as a reference helps the reader and the analyses minimize the measurement errors and facilitates understanding of the results. We have also clarified this in Section 2.1.2 as the Reviewer recommended by adding the following statement (lines 194-196):
‘The software converts inputs of food and drinks intakes into quantified macro- and micro-nutrient intakes and creates a mean of the two dietary recalls.’
(10) Tables 4 and 5: There are a lot of interesting details in these tables, thanks. It may be helpful if you can clearly note for the reader in the table or the footnote that the beta coefficient is associated with HR ethnicity.
We have added a brief explanation in the footnotes of both tables, that informs readers that beta coefficients are associated with HR ethnicity as follows (lines 301-302 and lines 323-324):
‘…and the β coefficient is associated with HR ethnicity.’
(11) Tables 4 and 5: The referenced nutrient recommendations do not always seem to apply to all age or sex groups. For example it looks like the WHO recommended intakes for vitamin A in women aged 19-65 is listed as 500 ug RE in reference #75 but you have only referenced 600 ug RE (the recommended intake for men, and women 65+). When there are multiple reference values that apply to your study population (i.e. men and women aged 20 to 64 years) it would be helpful to list the different values. In the Hungarian General and Hungarian Roma columns it would also be helpful to give male and female mean intakes for each nutrient, in the main text or in a supplement (even if a nutrient has the same requirement among males and females, it seems awkward to compare intakes between the general and HR populations because the proportion of females in the HR group is much higher), and it would also be interesting to know if there were important male – female differences (statistical tests not needed).
We thank the Reviewer for the in-depth review and the detailed recommendations. Regarding the WHO recommended value for Vitamin A, we used the 500 μg RE, value and in Table 5, but the ‘600 μg RE’ value is a typo, which we have corrected for the reference 70 in the revised version. In addition to help the interested reader and to address Reviewer’s suggestion, we have added two additional tables in the supplementary material (Table S3 and S4) in which we give detailed information on the macro- and micro-nutrient intakes among Hungarian Roma and general populations compared by sex. We have observed no major differences among males and females. Additionally, we have inserted the following statement in the manuscript to orient the interested reader to the new Tables S3-S4 (lines 332-333):
‘For more in-depth comparisons of nutrient intakes between males and females see Tables S3-S4.’
(12) Figure 1: When you assigned the outcome variable (nutrient adequacy) to individuals, did you only use the reference values listed in Tables 4 and 5, or did you use age and sex-specific values (when they were different across groups)? Even though you adjusted for gender, the analysis will not be correct unless you have accurately assigned adequacy at the individual level.
Apart from the limitations, which we have mentioned and expanded in Section 4.1 ‘Strengths, limitations and considerations for future research’, during all the steps of our analysis we have tried to adequately address adequacy at individual level and used age- and sex-specific, where relevant, in order to allow for more fine-tuned and accurate comparisons. More specifically, in Figure 1, our calculations are based on age- and sex-specific nutrient recommendations according to WHO, where applicable. In order to emphasize this to the interested reader, we have added a footnote in Figure 1 as follows (line 344):
‘Calculations are based on age and sex-specific nutrient recommendations according to WHO, where applicable.’
In addition, as explained earlier, we have been very specific for our software, which provides in-depth nutritional data for the Hungarian context and with Hungarian dishes and recipes, which is an advantage that we believe strengthens the quality of our data and analyses.
(13) Tables 4 and 5, and Figure 1: When estimating the beta coefficients in tables 4 and 5, and the odds ratios in Figure 1, did you consider or examine potential interactions between ethnicity and other variables that you controlled for (age, gender, education, marriage, financial status)?
(14) The discussion is well written. In Section 4.1 Paragraph 1: Can you please provide some general information from other studies or that you otherwise know of regarding diet during months not covered by your survey (September to April), and between northeast HR and other assimilated HR (this would be helpful to help contextualize the limitations that you have mentioned)?
To address this, we have added an extensive elaboration of nutritional studies involving Roma in different periods, locations and with different methods in order to contextualize the limitations we have mentioned earlier. A new part added in the Discussion section describes that independently when, where and what method used the Roma dietary quality was found persistently inferior compared to that of non-Roma populations. Description can be found as follows (lines 469-502):
‘Previous research on Roma attempting to characterize eating habits and food choices during different periods of the year than the current research (April - September), as well as different locations where Roma reside, have also recorded an unfavorable dietary picture. Roma youth from Slovakia were reported to consume fruits less frequently than non-Roma individuals [21] Roma participants (over 18 years of age) in survey from South Bohemian Region during June 2015 to March 2016, reported high consumption of sugar sweetened beverages and inadequate consumption of fruits and vegetables [24]. Another survey involving Roma and non-Roma subjects, conducted in the latter region, also reported low fruits and vegetables consumption [31]. A report from a European project involving Roma individuals from seven different European countries (i.e. Bulgaria, Czech Republic, Greece, Portugal, Romania, Slovakia, Spain) conducted in varying periods of the year (from March – June and September – December 2008) found that less than a third of Roma participants reported consuming fruits and fresh vegetables on a daily basis, while 36% reported consuming sweets and confectionery every day [26]. Research involving Roma from the county of Rimavská Sobota district of Slovakia (the survey was carried out during June 2007 – May 2008) showed that their diet was characterized by low consumption of dairy products and vegetables [27]. A cross-sectional study over a one-year period in 2010, using a questionnaire n instrument with questions related to the frequency of consumption for certain food groups among 400 participants from Roma communities in five districts in Albania, found infrequent consumption of fruits and milk and moderate consumption of meat and vegetables [28]. Further, estimates of the Household Budget Survey in Romania involving almost 9,000 settlements during 2004 to 2011, suggest that the Roma population has an inferior diet compared to that of non-Roma populations in terms of a lower proportion of dairy product, fruit, and vegetable intake [29]. Other studies involving Roma participants also suggest excessive consumption of fast-food, fatty meats and sweets and low consumption of vegetables and fruits [30] [25]. Eventually, findings from two paired health surveys, that we carried out in the general Hungarian and Roma populations using the same methodology before and after the Decade of Roma Inclusion were compared, and it was clearly shown that the distribution of BMI worsened significantly among younger Roma individuals (in both sexes) between 2004 and 2015, with obesity becoming significantly more frequent [3]. Such findings support the assumption that unhealthy diet is characteristic of the nutritional profile of the Hungarian Roma population, since obesity is a potential consequence of poor dietary behavior [100]. Despite the fact that all the above-mentioned studies have not quantified intake of micro- and macro-nutrients among Roma, it appears that dietary quality of Roma is less favorable than that of the host population, regardless of where their live or when the dietary survey was conducted. Our results are in line with these findings.’
References
- Kósa Z, Moravcsik-Kornyicki Á, Diószegi J, Roberts B, Szabó Z, Sándor J, et al. Prevalence of metabolic syndrome among Roma: a comparative health examination survey in Hungary. Eur J Public Health. 2014;25(2):299-304. doi: doi.org/10.1093/eurpub/cku157.
- Macejova Z, Kristian P, Janicko M, Halanova M, Drazilova S, Antolova D, et al. The Roma Population Living in Segregated Settlements in Eastern Slovakia Has a Higher Prevalence of Metabolic Syndrome, Kidney Disease, Viral Hepatitis B and E, and Some Parasitic Diseases Compared to the Majority Population. International Journal of Environmental Research and Public Health. 2020;17(9):3112. doi: 10.3390/ijerph17093112.

Round 2
Reviewer 1 Report
Thank you for taking into account my suggestions.
Reviewer 3 Report
Thank you very much for your thoughtful and detailed revisions which have fully addressed my comments. Many congratulations on preparing this excellent and important paper on the diet and nutrition status of this interesting population.